# Bacterial Community Shifts Driven by Nitrogen Pollution in River Sediments of a Highly Urbanized City

**DOI:** 10.3390/ijerph16203794

**Published:** 2019-10-09

**Authors:** Xianbiao Lin, Dengzhou Gao, Kaijun Lu, Xiaofei Li

**Affiliations:** 1Laboratory of Microbial Ecology and Matter Cycles, School of Marine Sciences, Sun Yat-Sen University, Zhuhai 519082, China; linxb7@mail.sysu.edu.cn; 2School of Geographic Sciences, East China Normal University, Shanghai 200241, China; gaodz0526@163.com; 3Southern Laboratory of Ocean Science and Engineering (Guangdong, Zhuhai), Zhuhai 519000, China; 4The University of Texas at Austin Marine Science Institute, 750 Channel View Drive, Port Aransas, TX 78373, USA; kaijun.lu@utexas.edu; 5Key Laboratory for Humid Subtropical Eco-geographical Processes of the Ministry of Education & School of Geographical Sciences, Fujian Normal University, Fuzhou 350007, China

**Keywords:** bacterial community, nitrogen pollution, river sediment, urbanization

## Abstract

Effects of nitrogen pollution on bacterial community shifts in river sediments remain barely understood. Here, we investigated the bacterial communities in sediments of urban and suburban rivers in a highly urbanized city, Shanghai. Sediment nitrate (NO_3_^−^) and ammonia (NH_4_^+^) were highly accumulated in urban river. Operation Taxonomic Units (OTUs), Abundance-based Coverage Estimators (ACEs) and Chao 1 estimator in urban rivers were slightly lower than those in suburban rivers, while Shannon and Simpson indices were higher in urban rivers than those in suburban rivers. *Proteobacteria*, *Firmicutes*, and *Bacteroidetes* were the dominant bacterial phylum communities, accounting for 68.5–84.9% of all communities. In particular, the relative abundances of *Firmicutes* and *Nitrospirae* were significantly higher in suburban rivers than in urban rivers, while relative abundances of *Bacteroidetes, Verrucomicrobia,* and *Spirochaetes* were significantly lower in suburban rivers than in urban rivers. NH_4_^+^ was significantly and negatively correlated with abundances of *Firmicutes*, *Nitrospirae*, and *Actinobacteria*. Importantly, the significant and negative effects of sediment NH_4_^+^ on bacterial richness and diversity suggested that nitrogen pollution likely contribute to the decrease in the bacterial richness and diversity. The results highlight that nitrogen enrichment could drive the shifts of bacterial abundance and diversity in the urban river sediments where are strongly influenced by human activities under the rapid urbanization stress.

## 1. Introduction

Global nitrogen pollution is of increasing concern because serious nitrogen pollution has a negative influence on the ecological environment [1,2]. A heavy nitrogen load is one of the pivotal drivers that contributes to water degradation and eutrophication in aquatic environments [3]. Due to the rapidly increasing population in urban areas, urban rivers receive a great amount of anthropogenic nitrogen, such as ammonia (NH_4_^+^), nitrate (NO_3_^−^), and nitrite (NO_2_^−^), which has become an important environmental problem [4]. In addition, urban rivers, as a result of extensive nitrogen input, are considered significant regional nitrous oxide (N_2_O) source hotspots that contribute to a part of global greenhouse effects [5]. Therefore, studies regarding nitrogen pollution and associated ecological effects have been going on for decades. Numerous studies have documented that urbanization has a great influence on nitrogen pollution in urban rivers [6,7,8]. In urban regions, large amounts of nitrogen from vehicle emission and industrial activity releases can be finally transported into the rivers by surface runoff and sewage discharge [9,10]. Importantly, household waste water discharge is also the main source of external NH_4_^+^, accounting for roughly 50% of total nitrogen input in central urban rivers [9,11].

It has been suggested that environmental factors affect microbial richness and diversity in urban rivers ecosystems [6,10,12,13]. Zhang et al. [14] reported that total phosphorus, pH, and NH_4_^+^ were significant properties affecting microbial community composition. Additionally, *Actinobacteria* and *Betaproteobacteria* taxa are strongly correlated with organic carbon, while chemolithoautotrophic bacterial community is highly abundant in the low oxygen environments [15]. It has been suggested that changes in C/N ratio and dissolved oxygen (DO) drive the shifts of microbial community composition [16]. High NH_4_^+^ content is favorable for the *Nitrospira*, which has also been identified in the urban river sediments [6]. Nitrogen pollution in urban rivers is heavily affected by the rapid urban development [11]. In highly urbanized city, the domestic wastewater discharge is the major source of nitrogen (especially NH_4_^+^) in the urban rivers [9,13]. A previous study reported that NH_4_^+^ has a positive effect on *Actinobacteria* abundance, and NO_3_^−^ shows a positive correlation with *Armatimonadetes*, *Chloroflexi* and *Bacteroidetes* abundances [14]. In addition, nitrogen transformation pathways in urban rivers have been reported to alter in response to the increasing nitrogen input from anthropogenic activities [17,18], which may be attributed to the shifts of bacterial community [6,19]. Although these studies have improved our understanding of the alterations in microbial community composition and nitrogen cycling under nitrogen pollution stress, knowledge of the dynamics of bacterial diversity and abundance in river sediments impacted by large amount of nitrogen inputs is still limited in the highly urbanized areas. Specifically, it has been reported that NH_4_^+^ is highly accumulated in the sediments, playing an important role of nitrogen source and sink for overlying water [9,11,14]. However, the effects of sediment NH_4_^+^ on the bacterial richness and diversity remain unclear in urban rivers.

The river networks in Shanghai city are highly developed, with a water area of 569.6 km^2^ crisscrossing this city, accounting for about 9% of total land areas (6340 km^2^). With the rapid urbanization of Shanghai city, the river networks have suffered heavy eutrophication and algal blooms due to excessive external nitrogen and phosphorus inputs from household and industrial sewage discharge [5,11]. The water bodies and sediments are black and smelly (Appendix A), which has attracted government attention since the degradation of both water quality and ecological function in urban river systems is attributed to nitrogen pollution. Urbanization has a great influence on nitrogen pollution in the urban rivers because most source of nitrogen derives from human activities [11,13]. The central urban area of Shanghai accounts for 40% of the total population, contributing to the serious nitrogen pollution in the central urban area [17]. The concentration of nitrogen in river sediments is significantly higher in the central urban area than in suburban areas in Shanghai city [5,17]. To date, most previous studies have been concerned with N_2_O emissions and nitrogen load [5,11], and there is limited information on the effect of nitrogen pollution on bacterial dynamics in the river sediments of the highly urbanized city of Shanghai. Therefore, it is necessary to examine the shifts in microbial community induced by the nitrogen pollution in urban rivers, which will contribute to our knowledge of its environmental implications under a rapid urbanization scenario.

Therefore, we investigated the nitrogen pollution and dynamics of bacterial diversity and community structure in the river sediments of highly urbanized city, Shanghai. Thus, the objectives of our study were: (1) to examine whether shifts in the bacterial communities induced by nitrogen load had occurred, (2) to study which variables played a crucial role in shaping bacterial diversity and richness, and (3) to identify that which particular bacterial community was potentially resilient to the serious nitrogen pollution.

## 2. Materials and Methods

### 2.1. Study Area and Samples Collection

Located in the easternmost region of the Yangtze River Delta of China, Shanghai is the most economically advanced and also the most densely populated city in China. In 2016, the Gross Domestic Product (GDP) and population density of Shanghai were 2817.87 billion RMB and 3816 persons per square kilometer, respectively [20]. Recently, the river networks have suffered serious nitrogen pollution due to the large amount of domestic and industrial wastewater discharge [17]. Nitrogen pollution in the rivers has become the main environmental problem because excess nitrogen load can lead to the degradation of water quality. In this study, the sampling sites were located at central urban area (U1–U6) and suburban area (S1–S6) (Figure 1). Sampling surveys of sediment and overlying water samples were conducted on 10–15 January 2015. At each site, triplicate sediment cores were sampled by a Sediment Sampler (BWT2-04.23.SA, Beeker, Waterland International Co., Ltd., Holland) and the surface sediment sample (0–5 cm depth) was collected. Overlying water was collected by a Ruttner Water Sampler (KC-denmark, Silkeborg, Denmark). Sediment and overlying water samples were stored into sterilized 50 mL centrifuge tubes and 500 mL polyethylene bottles, respectively. These samples were placed on ice and transported into laboratory within 4 h. pH and DO of overlying water were respectively measured by a pH Meter (Mettler-Toledo, Columbus, OH, USA) and a HQ 40dm portable water quality analyzer (HACH, Loveland, CO, USA) during the field investigation. Upon return to the laboratory, each sediment sample was homogenized thoroughly under helium and divided into two fractions. The first fraction was stored at –80 °C for molecular analyses, and the second fraction was stored at 4 °C for measurements of sediment properties.

### 2.2. Analysis of Bottom Water and Sediment Characteristics

Bottom water samples were filtered through 0.45 μm cellulose acetate filters. NH_4_^+^, NO_3_^−^ and NO_2_^−^ in bottom water were determined via a continuous-flow nutrient autoanalyzer (SAN plus, Skalar Analytical B.V., Breda, The Netherlands) with detection limits of 0.1 μmol L^–1^ for NO_2_^−^/NO_3_^−^ and 0.5 μmol L^–1^ for NH_4_^+^ [21]. Dissolved organic carbon (DOC) in bottom water was measured by a total organic carbon analyzer (TOC-V CPH). Total organic carbon (TOC) and total nitrogen (TN) in sediments were determined using a thermal combustion furnace analyzer (Elementar analyzer vario MaxCNOHS, Frankfurt, Germany). NH_4_^+^ and NO_3_^−^ in sediments were extracted with 2 mol L^–1^ KCl solution at the speed of 250 r min^-1^ shaking for 60 min and measured using a nutrient autoanalyzer [21]. Sulfide in sediment was determined by the method of methylene blue spectrophotometry. Ferrous oxides (Fe(II)) in sediment was extracted by 10 mL 0.5 mol L^–1^ HCl, and measured using the ferrozine method [22].

### 2.3. DNA Extraction, Amplification and 16S rRNA Gene-Based Pyrosequencing

DNA in sediment was extracted from 0.25 g fresh sediment using Power Soil DNA Isolation Kits (MOBIO, Carlsbad, CA, USA) according to the manufacturer’s protocols. Purity and content of DNA were measured using a Nanodrop-2000 Spectrophotometer (Thermo Fisher Scientific, Waltham, MA, USA). The V4 hypervariable region of bacterial 16S rRNA gene was amplified using the primers F515 (5′-GTGCCAGCMGCCGCGGTAA-3′) and R806 (5′-GGACTACHVGGGTWTCTAAT-3′). Amplification of polymerase chain reaction (PCR) were performed in 25 μL reaction mixtures including 1.0 μL of each primer (10 μM), 1 μL of template DNA, 12.5 μL of Taq PCR Master Mix (2×, with blue dye), and 9.5 μL of sterile distilled water. The thermos cycling was conducted in the condition of 98 °C for 1 min followed by 35 cycles of 10 s at 98 °C, 30 s at 50 °C, 30 s at 72 °C, and a final 5 min extension cycle at 72 °C. PCR amplicons were purified using Agarose Gel DNA Purification Kit (TaKaRa Biotechnology, Dalian, China) and quantified using a Quant-It Pico Green kit (Invitrogen, Carlsbad, CA, USA) with a Qubit fluorometer (Life Technologies, Carlsbad, CA USA). Subsequently, the purified amplicons were submitted to Novogene Beijing for Illumina paired-end (PE) library preparation, cluster generation and 250 bp PE sequencing on an Illumina MiSeq machine.

### 2.4. Processing of Sequence Data

In this study, the raw sequence data was analyzed with Quantitative Insights into Microbial Ecology (QIIME) toolkit (version 1.9.0, http://bio.cug.edu.cn/qiime/). Raw FASTQ files were de-multiplexed with QIIME, and the paired reads were joined with FLASH (fast length adjustment of short reads) using default setting [23]. Sequences with low quality reads, unknown nucleotides and ambiguous reads and trimming off the barcodes and primers were removed from the dataset, and were subsequently used for the following analysis. The joined pairs were then filtered and analyzed with QIIME. Shannon diversity index [24] and Chao1 estimator [25] were calculated using the software MOTHUR. Sequences were clustered into operational taxonomic units (OTUs) based on 97% sequence identity by using the average neighbor algorithm. Rarefaction curves were drawn to compare OTU-based bacterial diversity among the measured samples of studied sites. The differences in bacterial community structure between urban rivers and suburban rivers were analyzed using a phylogeny-based weighted UniFrac distance metric. Heat map of dominated bacterial community composition at the genus level was drawn using R software (version 3.2.3, Lincoln, NE, USA). In addition, the abundance-based coverage estimators (ACE), and Simpson diversity indices of bacterial communities were calculated using MOTHUR software (version 1.22.2, The University of Michigan, Ann Arbor, MI, USA).

### 2.5. Statistical Analysis

Statistical analyses of this study were performed using SPSS (version 19.0, SPSS Inc., Chicago, IL, USA), and *p* < 0.05 values were considered to be significant. A one-way analysis of variance (ANOVA) with Tukey’s HSD test was conducted. The significant differences in bacterial richness and diversity between urban and suburban rivers were identified by the independent sample analysis with *t*-test. Pearson’s correlation analysis was also performed to explore the relationships of relative bacterial abundance and diversity indices and sediment and water variables. The overall difference in bacterial community structure was identified by nonmetric multidimensional scaling (NMDS) with weighted UniFrac NMDS plots. The difference of bacterial structure was identified using analysis of similarity (ANOSIM). Redundancy analysis (RDA) combined with Monte Carlo test (999 permutations) was performed to elucidate the relationships between bacterial richness and diversity and environmental variables using CANOCO software (version 4.5, Wageningen, The Netherlands).

## 3. Results

### 3.1. Chemical Characteristics of Overlying Water and Sediment

The chemical properties for sediments and overlying water are given in Table 1. The pH for overlying water ranged from 6.8 to 7.1 in urban rivers and from 7.2 to 7.6 in suburban rivers. DO in overlying water varied between 1.15 and 4.92 mg L^–1^, and DO was slightly lower in urban rivers than in suburban rivers. The contents of DOC in overlying water were 7.2–21.8 mg C L^−1^ and 5.8–19.6 mg C L^−1^ in urban rivers and suburban rivers, respectively. The contents of NO_3_^–^ and NO_2_^–^ in overlying water varied between 2.64 and 9.06 mg N L^–1^, and between 1.89 and 197 μg N L^–1^, respectively. However, NH_4_^+^ in overlying water varied spatially among the sample sites, with contents of 6.43–13.7 mg N L^–1^ in urban rivers and 0.01–0.24 mg N L^–1^ in suburban rivers. NH_4_^+^ in overlying water was significantly higher in urban rivers than in suburban rivers (*t*-test, *p* < 0.05, Table 1). Sediment sulfide was significantly higher in suburban rivers than in urban rivers (*t*-test, *p* < 0.05). Contents of sediment TOC and TN were 10.8–33.2 mg C g^–1^ and 1.35–3.26 mg N g^–1^, respectively. The contents of NO_3_^–^ and NH_4_^+^ in sediment ranged from 0.32 to 3.25 μg N g^–1^ and from 4.79 to 240 μg N g^–1^, and the contents were significantly higher in urban rivers than in suburban rivers (*p* < 0.05, Table 1, Appendix A). Sediment Fe(II) ranged from 7.89 to 15.4 mg Fe g^–1^ in urban rivers and from 0.43 to 25.1 mg Fe g^–1^ in suburban rivers. Contents of NH_4_^+^ in overlying water and sediments were significantly higher in urban rivers than in suburban rivers (*t*-test, *p* < 0.05, Table 1).

### 3.2. Bacterial Richness and Diversity Indexes

In this study, the 16S rRNA gene sequences ranged from 76,364 to 91,891 for each sample, with a total of 120,365 sequences from the 12 samples. The bacterial richness and diversity indexes were given in Table 2. The sequences were grouped into 51,505 OTUs based on 97% similarity. 3870–4762 OTUs of individual sample were observed, contributing to 7.51–9.24% of the total OTUs. ACE estimated by the abundance-based coverage was in the range of 4545–5455 and 4644–11,205 in urban and suburban rivers, respectively. The Chao 1 estimator of the species richness of community ranged from 4422 to 11,205. However, there was no significant difference in Chao 1 estimator between urban and suburban rivers (*t*-test, *p* > 0.05) (Appendix A). Likewise, the Shannon index indicating overall diversity of community varied between 8.0 and 9.8, and S5 had the lowest Shannon index and U6 had the highest Shannon index (Table 2). Likewise, the Simpson index ranged from 0.95 to 1.00, and were generally higher in urban rivers than suburban rivers. The coverage varied between 98.5% and 98.8% in urban rivers, and between 97.4% and 98.7% in suburban rivers, indicating a good coverage estimated in this study. Generally, the OTUs, ACE and Chao 1 estimator of the bacterial richness indexes were slightly lower in urban rivers than in suburban rivers. However, Shannon and Simpson of bacterial diversity were higher in urban rivers than in suburban rivers (Appendix A).

### 3.3. Phylogenetic Affiliation of 16S rRNA Gene Sequences

*Proteobacteria* was the most abundant phylum of the bacterial communities, accounting for 48.85–76.74% of the total bacterial sequences. The *Firmicutes*, *Bacteroidetes*, *Chloroflexi*, *Actinobacteria*, and *Acidobacteria* were also the dominated bacterial communities, and their relative abundances accounted for 3.65–27.65%, 2.21–10.63%, 3.19–8.33%, 1.34–5.27%, and 2.20–5.21% of total sequences, respectively. The relative abundances of *Firmicutes* and *Nitrospirae* were significantly higher in suburban rivers than in urban rivers (*t*-test, *p* < 0.05) (Appendix A, Figure 2). Mean relative abundances indicated that *Bacteroidetes, Verrucomicrobia,* and *Spirochaetes* were more abundant in urban rivers than in suburban rivers. On the contrary, the relative abundances of *Chloroflexi, Actinobacteria, Acidobacteria,* and *Gemmatimonadetes* increased from 5.12% (urban rivers) to 5.38% (suburban rivers), 2.40% to 2.72%, 3.50% to 3.63%, and 0.73% to 0.82%, respectively. The abundant bacterial communities of *Anaeromyxobacter, Burkholderia, Sulfurimonas,* and *Novosphingobium* at the genus level were detected in urban rivers, while relatively high abundant *Sphingopyxis, Polaromonas, Perlucidibaca, Flavobacterium,* and *Acinetobacter* were observed in suburban rivers (Figure 3).

At the family level, bacterial sequences were assigned to 392 families. The top 36 abundant families accounted for 50.7–66.8% of total bacterial sequences. 25 families assigned to phylum *Proteobacteria* accounted for 69.4% of the top families (Appendix A). *Rhodocyclaceae* (6.26–12.96%), *Comamonadaceae* (2.60–6.08%), *Gallionellaceae* (2.11–4.74%), *Geobacteraceae* (2.07–3.96%) and *Hydrogenophilaceae* (1.68–3.73%) were also the most abundant communities in urban rivers, while *Rhodocyclaceae* (4.17–6.97%), *Comamonadaceae* (2.33–5.43%), *Anaerolineaceae* (2.34–4.38%), *Gallionellaceae* (2.11–4.74%), *Planococcaceae* (2.11–8.85%), and *Xanthomonadaceae* (1.60–3.30%) contributed substantially to the bacterial communities in suburban rivers.

However, the bacterial community was dominated by the family *Bacillaceae* at sites S4 and S5, accounting for 16.63% and 21.33%, respectively. *WCHB1-69*, *Syntrophaceae*, *Rhodocyclaceae*, *Geobacteraceae*, *unidentified_Xanthomonadales*, and *Spongiibacteraceae* were significantly higher in urban rivers than in suburban rivers (*t*-test, *p* < 0.05), while only relative abundance of *Planococcaceae* (affiliated to phylum *Firmicutes*) was significantly lower in urban rivers than in suburban rivers (*t*-test, *p* < 0.05).

### 3.4. Factors Shaping Bacterial Community Structure and Diversity

The result of nonmetric multidimensional scaling ordination analysis indicated a differentiation in the bacterial communities between urban and suburban rivers (stress = 0.073, Figure 4). Bacterial community structure was further evidenced by an analysis of similarity (ANOSIM), which revealed that the site condition may be the important determinants of bacterial community composition (Appendix A). The Pearson’s correlation analysis indicated that ACE and Chao 1 estimator were significantly and positively related with DO, while negatively with Fe(II) (*t*-test, *p* < 0.05; Table 3). In addition, significantly negative relationships were observed between sediment NH_4_^+^ and OTUs, ACE and Chao 1 estimator (*p* < 0.05). The Pearson’ analysis revealed that NH_4_^+^ and NO_3_^–^ were the main properties that significantly affected the abundance of phylum *Firmicutes*, *Nitrospirae*, *Verrucomicrobia* (*p* < 0.05; Appendix A). In addition, the scatter plot of NMDS indicated strong differences in the bacterial communities between urban and suburban rives, with most urban rivers clustering on the left side of the y-axis and suburban rivers clustering on the right side of the y-axis (Figure 4).

### 3.5. Redundancy Analysis

Redundancy analysis (RDA) was performed to reveal the relationships between bacterial richness and diversity and environmental variables. Ordination triplets of first two axes (RDA1 and RDA2) explained 73.2% of the total variance (Figure 5). Results of RDA indicated that bacterial richness and diversity were significantly correlated with overlying water and sediment parameters. The first axis of RDA1 was positively correlated with overlying water DO, pH and NO_3_^–^, sediment sulfide, but negatively correlated with sediment NH_4_^+^, water NH_4_^+^ and C/N, which explained 48.7% of the total variance (*p* < 0.05, Monte Carlo based on 999 permutations). The second axis of RDA2 was positively correlated with TOC and TN, negatively related to water NO_2_^−^ and sediment NO_3_^−^, explaining 24.5% of the total variance (*p* < 0.05, Monte Carlo based on 999 permutations). DO had a significantly positive effect on Chao 1 estimator and ACE (*p* < 0.05, based on 999 permutations). Sediment NH_4_^+^ showed a significantly negative influence on OTUs, ACE and Chao 1 estimator (*p* < 0.05, based on 999 permutations).

## 4. Discussion

Urban rivers in rapid urbanization areas are easily affected by the land use and human activities [6,26]. Subsequently, the chemical properties and ecological function of overlying water and sediment in urban rivers are susceptible to anthropogenic pollutant inputs [6,10]. The main source of nitrogen pollution in urban rivers is household and industrial sewage discharge [17]. In addition, the content of reactive nitrogen in urban rivers increases intensively at a rapid rate under urbanization [11]. Therefore, shifts in the bacterial community composition and diversity are likely to occur in response to the increasing nitrogen pollution in urban rivers [27,28]. In this study, the nitrogen enrichment in the urban rivers could contribute to bacterial community shifts in sediments. Also, particular bacterial communities are potentially resilient to nitrogen pollution.

Previous studies reported that the dominant bacterial phyla communities in urban rivers were *Proteobacteria*, *Firmicutes*, *Actinobacteria*, *Bacteroidetes*, and *Spirochaetes*, and their relative abundances were related to fecal pollution [8,28]. In addition, it has been reported that the bacterial phyla in the urban rivers of Nanjing city were dominated by *Proteobacteria*, *Actinobacteria*, *Bacteroidetes*, following by *Verrucomicrobia*, *Chlorobi*, *Firmicutes*, *Planctomycetes*, *Cyanobacteria* and *Chloroflexi* [28]. However, the bacterial community differed with a season variation and by sample site, indicating that the environmental properties strongly regulate the dynamics of bacterial communities in urban rivers [28]. A recent study has documented that nitrogen addition can increase the relative abundances of *Proteobacteria* and *Actinobacteria*, but decrease the relative abundances of *Acidobacteria*, *Verrucomicrobia*, *Planctomycetes*, and *WD272* [29]. Importantly, the relative abundances of *Burkholderia* and *Rhizomicrobium* for nitrogen cycling bacterial communities increased with the nitrogen addition [29]. Likewise, our study also indicated that a high abundance of *Actinobacteria* was observed in the high NH_4_^+^ content sites of the urban rivers (Figure 2). The bacterial communities in sediments of the Upper Mississippi River were dominated by *Proteobacteria*, *Bacteroidetes*, *Acidobacteria*, and *Actinobacteria*, and these bacterial communities varied temporally and spatially [30]. In our study, *Firmicutes*, *Bacteroidetes*, *Chloroflexi*, *Actinobacteria*, and *Acidobacteria* were dominant in the river sediments. However, the relative abundances of *Bacteroidetes, Verrucomicrobia,* and *Spirochaetes* were higher in urban rivers than in suburban rivers, while the relative abundances of *Chloroflexi, Actinobacteria, Acidobacteria,* and *Gemmatimonadetes* were higher in suburban rivers than in urban rivers [8]. In an eutrophic lake, NO_3_^–^ content level dominated the bacterial community composition [31]. However, the bacterial community was dominated by *Actinobacteria*, *Beta-* and *Alphaproteobacteria* in the sediments of the nitrogen-enriched lake [31]. The difference in the bacterial community composition between urban and suburban rives could be partly attributed to the important effects of sediment NH_4_^+^ on the bacterial communities.

The significant difference in chemical properties between urban and suburban rivers was the inorganic nitrogen content, which may play a crucial role in the special bacterial variations. NH_4_^+^ in overlying water and NH_4_^+^/NO_3_^–^ in sediments were significantly higher in urban rivers than in suburban rivers (Table 1). Domestic wastewaters and human excreta contribute the most to NH_4_^+^ in urban rivers [9,11,32,33]. Therefore, the NH_4_^+^ content likely contributes to the abundance of bacterial communities and variations of special communities. The OTUs, ACE, Chao 1 estimator in this study were negatively related to sediment NH_4_^+^ content (Table 3), indicating that the heavy NH_4_^+^ pollution reduces the bacterial richness in urban rivers. Drury et al. reported that NH_4_^+^ enrichment in urban sediment supply the substrate for nitrification process [6]. The abundance of *Nitrospirae* was higher in urban river sediment than in suburban river sediment because *Nitrospira* species are nitrifying Gram-negative chemoautotrophic bacteria [12]. The relative abundances of *Firmicutes* and *Nitrospirae* were lower in suburban rivers with NH_4_^+^ enrichment, suggesting that excessive NH_4_^+^ can inhibit these two bacterial communities although they are involved in the fecal pollution and nitrogen cycling. The relative abundance of *Nitrospirae* in this study was significantly higher in suburban rivers than in urban rivers (*t*-test, *p* < 0.05), which suggested that the serious NH_4_^+^ pollution had a negative influence on this bacterial community. Although the NH_4_^+^ contents (6.4–15.4 mg N L^–1^) are extremely high in the urban rivers, nitrifying bacteria cannot be directly inhibited by the toxicity of the NH_4_^+^ in this study due to that nitrifying bacteria can work with high NH_4_^+^ amount (up to 196 mg N L^–1^) [34]. Thus, the attribute in this result may be that the nitrifiers were inhibited by low DO concentration in the urban rivers, which was induced by excessive NH_4_^+^. Meanwhile, this result may be also attributed to other pollutants, such as heavy metals and permanent organic pollutants, as reported in previous studies in urban river ecosystems [35,36,37,38]. In addition, *Firmicutes* is the Gram-positive heterotrophic bacteria for nutrient cycling, the richness of *Firmicutes* was thus affected by the fecal pollution [8], and the *Firmicutes* community was the indicator of NH_4_^+^ and NO_3_^–^ derived from anthropogenic activities [38]. On the contrary, our results showed that more abundant *Firmicutes* was observed in suburban rivers compared to urban rivers, because excess amount of NH_4_^+^ pose a noxious action to *Firmicutes* in the high content of NH_4_^+^ in the urban rivers [39]. Therefore, NH_4_^+^ from anthropogenic activities in the urban rivers may contribute to the lower abundance of *Nitrospirae* and *Firmicutes* bacteria [6].

Due to the increasing nitrogen load in rivers, nitrogen makes a stronger contribution to microbial community dissimilarities than other variables in urban rivers [28]. It has been suggested that NH_4_^+^ was a critical environmental parameter determining the composition of the soil bacterial community [29]. In this study, the OTUs, ACE and Chao 1 estimator decreased with the increasing NH_4_^+^ content, suggesting that NH_4_^+^ pollution likely contributes to decrease in bacterial diversity. However, in the present study, sediment NH_4_^+^ had a negative influence on the bacterial diversity, suggesting the decrease in bacterial richness in response to the serious nitrogen pollution because only a few bacterial communities may survive in the environments impacted by the excessive nitrogen. The negative effect of NH_4_^+^ on bacterial richness was also evidenced in soil environments [40]. In addition, the occurrence of *Microcystis* can decrease the bacterial diversity since the eutrophication driven by the nitrogen [31]. In addition, sediment NH_4_^+^ was identified as the key driving factors of changes in the bacterial composition [17]. Therefore, the nitrogen pollution in rivers had both direct and indirect influences on the bacterial diversity. In addition, the positive correlation between DO and bacterial indices suggested that the high DO content was favorable for aerobic bacterial communities in the rivers.

In addition to the nitrogen, DO, pH and sulfide are also the important factors shaping the bacterial structure and composition in wide land and aquatic ecosystems [28,41]. The decrease in DO content may be attributed to the serious nitrogen pollution, resulting in the degradation of water quality in the rivers. It has been reported that DO has a considerable influence on the bacterial composition because the DO content level directly affects the organism metabolism [42]. In this study, the positive correlation between DO and both ACE and Chao 1 estimator further suggested the importance of DO on the regulation of the bacterial richness in the urban rivers. It has been reported that sulfide has a negative influence on microbial metabolism [43]. In this study, a negative correlation between ACE and sulfide further was also observed. In addition, temperature can contribute to the differences in bacterial phyla abundances, because relatively high temperature can promote the bacterial activity [30,44]. The chemical properties in overlying water were significantly correlated with those in the sediments (Appendix A), further indicating that properties of overlying water have considerable influences on the bacterial community structure and diversity (Table 1). Perryman et al. reported that the denitrifying community differs significantly between urban sediments and non-urban sediments because the nitrate is seriously polluted in urban sediment [27]. Local environmental conditions play a key role in shaping bacterial communities and the dynamics of environmental factors could be important in revealing bacterial distribution in the study area [45]. In this study, the local sites conditions and chemical properties varied between urban and suburban rivers, which likely resulted in the variations of the bacterial communities [38].

## 5. Conclusions

Our results indicate shifts in bacterial richness and diversity in response to the nitrogen pollution in highly urbanized river sediments. The overlying water NH_4_^+^ and sediment NH_4_^+^ and NO_3_^–^ was significantly higher in urban rivers than in suburban rivers, indicating that heavy human activities contribute to the NH_4_^+^ pollution in urban rivers. The significant decrease in the abundances of phyla *Firmicutes* and *Nitrospirae*, and the increase in phyla *Bacteroidetes*, *Verrucomicrobia, Spirochaetes* dominated the dynamics of bacterial community structure in response to the nitrogen pollution. The bacterial community shifts at the family level showed that relative abundances of *Syntrophaceae*, *Rhodocyclaceae*, *Geobacteraceae*, *unidentified_Xanthomonadales* and *Spongiibacteraceae* were higher in urban rivers in suburban rivers, which could be potentially beneficial to high nitrogen stress. NO_3_^–^ and NH_4_^+^ were significantly correlated with the abundances of *Firmicutes*, *Nitrospirae*, *Actinobacteria*. In addition, NO_3_^–^ and NH_4_^+^ had direct and indirect influences on the bacterial richness and diversity, suggesting the negative effects of nitrogen load on bacterial communities. Overall, the changes in bacterial structure and declining in bacterial richness and diversity were greatly affected by the nitrogen pollution. Furthermore, increasing nitrogen input to urban rivers caused by the rapid urbanization can affect not only the chemical properties, but also the bacterial community compositions, and may further alter the ecological importance of specific communities in the river environments.

## Figures and Tables

**Figure 1 ijerph-16-03794-f001:**
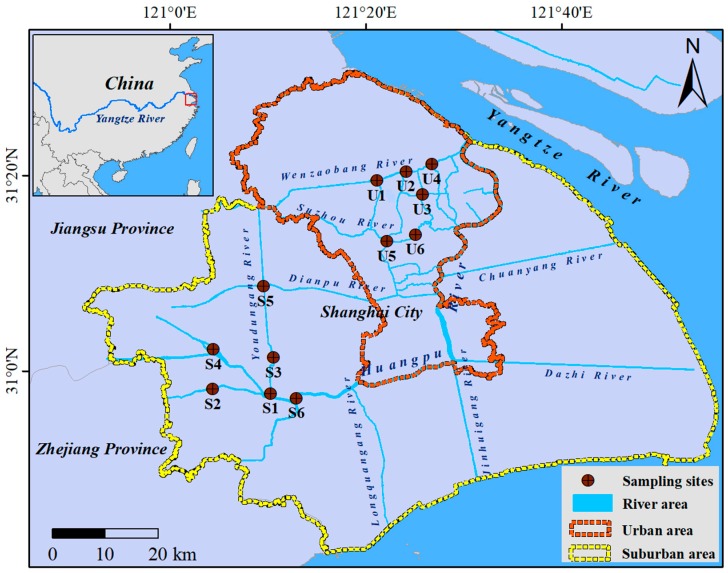
The sampling sites were located at the urban rivers (U1–U6) and suburban rivers (S1–S6) of the highly urbanized city (Shanghai, China).

**Figure 2 ijerph-16-03794-f002:**
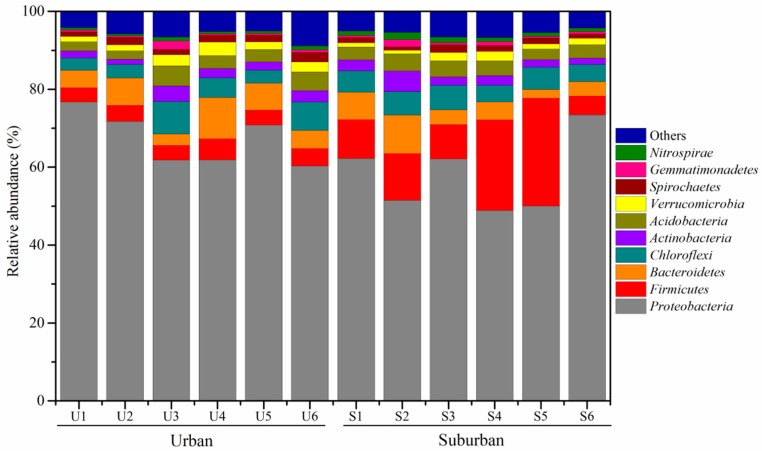
Relative abundance of sediment top 10 bacterial community compositions at phylum level. The relative abundance is expressed as the percentage of the targeted sequences to the total high-quality bacterial sequences of samples. “Others” refers to the taxa with a maximum abundance of <1% in any sample.

**Figure 3 ijerph-16-03794-f003:**
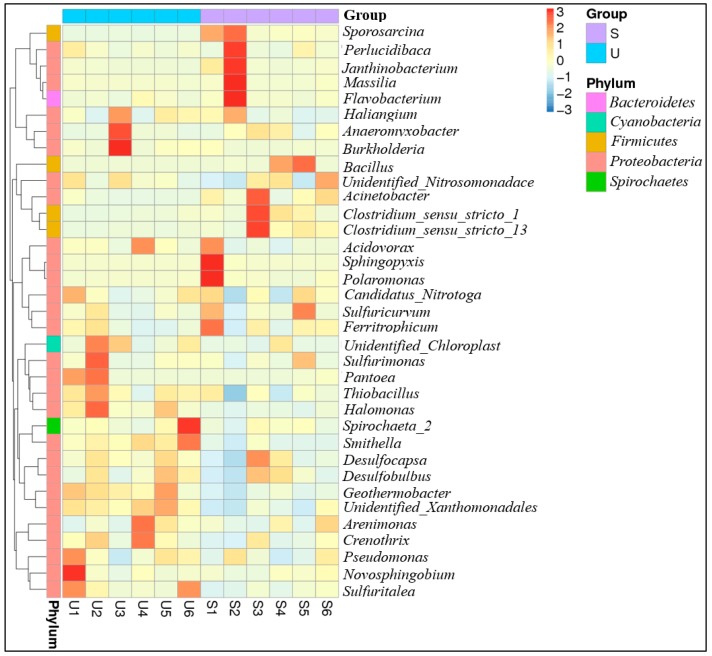
Heat map of top abundant genus level in each sample. The color intensity in each cell indicates the transformed relative abundance [log_2_ (x + 0.01)] of a family in a sample, referring to color key at the right top of the figure. The families in blue or red showed lower or higher relative abundances among samples.

**Figure 4 ijerph-16-03794-f004:**
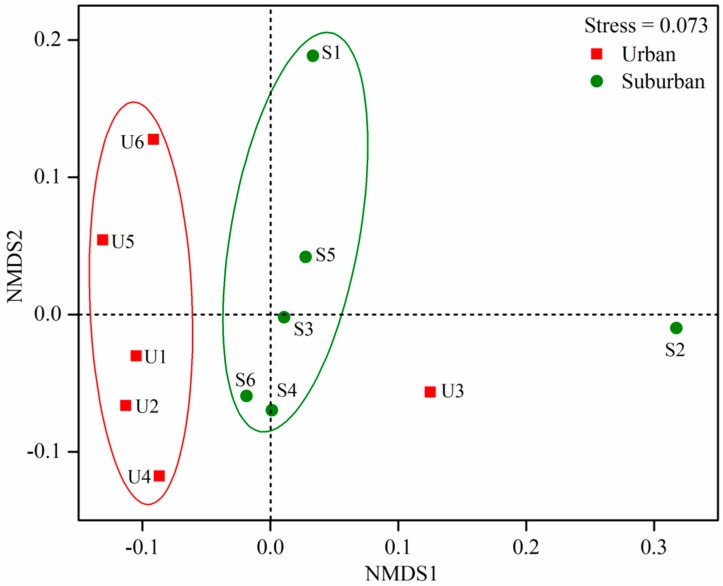
Nonmetric multidimensional scaling (NMDS) ordination of the dissimilarity (Bray-Curtis distance) in bacterial community composition.

**Figure 5 ijerph-16-03794-f005:**
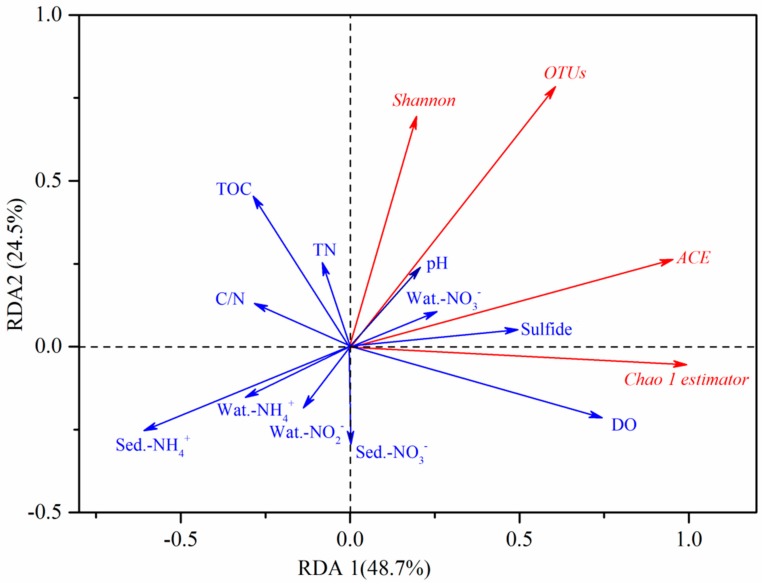
Redundancy analysis (RDA) compared to bacterial community richness and diversity and environmental properties. The percentages of total variation were explained by the first two axes are shown in parentheses.

**Table 1 ijerph-16-03794-t001:** Chemical properties of overlying water and sediments in the urban and suburban rivers.

Sites	Overlying Water						Sediment						
	pH	DO (mg L^–1^)	DOC (mg C L^–1^)	NO_3_^–^ (mg N L^–1^)	NH_4_^+^ (mg N L^–1^)	NO_2_^–^ (μg N L^–1^)	Sulfide (μg S g^–1^)	TOC (mg C g^–1^)	TN (mg N g^–1^)	C/N	NO_3_^–^ (μg N g^–1^)	NH_4_^+^ (μg N g^–1^)	Fe(II) (mg Fe g^–1^)
U1	6.9	2.89	21.8	3.81	15.4	8.03	158	10.8	1.51	7.17	2.25	188	9.87
U2	6.9	2.58	11.6	2.80	11.9	1.89	216	18.5	1.35	13.7	3.11	234	15.4
U3	7.0	2.68	14.4	3.39	13.7	89.5	152	33.2	3.15	10.5	2.93	147	7.89
U4	7.1	1.24	18.9	2.64	10.7	197	130	17.9	3.26	5.48	2.07	140	8.47
U5	6.8	2.01	13.6	9.06	6.43	12.1	4.15	22.1	2.43	9.12	3.25	240	9.83
U6	7.0	1.15	7.2	5.10	8.90	8.03	32.5	18.3	2.46	7.47	0.79	135	8.69
S1	7.3	2.65	19.6	3.29	0.24	5.47	515	15.3	1.06	14.4	0.32	38.8	9.01
S2	7.2	4.92	6.3	5.78	0.04	15.2	669	12.5	1.87	6.70	1.87	4.79	0.43
S3	7.3	1.31	5.8	2.77	0.02	12.1	274	22.8	3.11	7.34	1.35	52.5	9.68
S4	7.5	3.11	8.7	5.68	0.01	7.52	815	26.7	2.08	12.8	0.57	154	9.39
S5	7.2	2.01	7.7	4.15	0.02	17.3	172	26.4	2.03	13.0	0.38	115	8.66
S6	7.3	2.17	10.4	3.69	0.04	2.40	248	19.2	1.62	11.8	0.63	132	25.1
*p*	<0.001	0.33	0.14	0.83	<0.001	0.21	0.01	0.93	0.37	0.26	0.007	0.01	0.92

The data was the mean value of triplicate samples measured. *p* <0.05 indicated that the properties differed significantly between grouped urban (U1–U6) and suburban (S1–S6) rivers.

**Table 2 ijerph-16-03794-t002:** Bacterial richness and diversity characteristics of the bacterial sequences from pyrosequencing at each sample site.

Sites	OTUs ^a^	ACE ^b^	Chao 1 ^c^	Shannon ^d^	Simpson ^e^	Coverage ^f^
U1	3881	4799	4664	8.4	0.98	98.5
U2	3883	4624	4515	8.7	0.99	98.7
U3	4599	5423	5247	9.5	0.99	98.5
U4	3870	4545	4422	8.9	0.99	98.8
U5	4215	4987	4899	9.1	0.99	98.6
U6	4762	5455	5308	9.8	1.00	98.6
S1	4522	5288	5176	9.3	0.99	98.6
S2	4756	7021	11,205	9.0	0.99	97.4
S3	4468	5208	5121	9.2	0.99	98.6
S4	4398	5270	5126	8.7	0.97	98.5
S5	4137	4947	4787	8.0	0.95	98.5
S6	4014	4741	4644	8.3	0.97	98.7

^a^ Operation taxonomic units (97% similarity), ^b^ Abundance-based coverage estimators, ^c^ Richness estimate for an OTU definition, ^d^ Non-parametric Shannon diversity index. ^e^ The inverse Simpson index. ^f^ unit was (%).

**Table 3 ijerph-16-03794-t003:** Summary of Pearson’s correlation analysis between chemical properties and bacterial richness and diversity indexes.

Environmental Parameters	OTUs	ACE	Chao	Shannon	Simpson
pH	0.33	0.29	0.21	−0.17	−0.43
DO	0.26	0.69 *	0.76 *	−0.16	−0.08
NO_3_^− a^	0.26	0.31	0.28	0.11	−0.02
NH_4_^+ a^	−0.34	−0.35	−0.32	0.20	0.46
NO_2_^−^	−0.23	−0.21	−0.14	0.14	0.25
Fe(II)	−0.55	−0.66 *	−0.59 *	−0.39	−0.34
Sulfide	0.34	0.53	0.51	−0.10	−0.20
TOC	0.18	−0.15	−0.32	0.09	−0.25
TN	0.18	−0.02	−0.09	0.40	0.28
C/N	−0.10	−0.24	−0.31	−0.29	−0.46
NO_3_^− b^	−0.26	−0.08	0.02	0.17	0.49
NH_4_^+ b^	−0.60 *	−0.65 *	−0.60 *	−0.19	−0.03

^a^ indicates properties of overlying water, ^b^ indicates properties of sediment. * *p* < 0.05 and ** *p* < 0.01 indicate a significant correlation.

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
