# Peer review of "Bacterial Community Shifts Driven by Nitrogen Pollution in River Sediments of a Highly Urbanized City"

_ijerph, 2019, doi:10.3390/ijerph16203794_

Round 1

Reviewer 1 Report

The manuscript is a detailed study of microbial communities present in waters and sediments of rivers in urban and suburban areas of Shanghai.

The study correlate the presence of various forms of nitrogen present with the microbial community abundance e biodiversity, and draws the conclusions that some forms of nitrogen negatively affect microbial communities, and could drive the shifts of bacterial abundance and diversity.

The study is well set up and well conducted, but lacks some analysis and data that could invalidate some of the conclusions drawn.

The study, in fact, ascribe the reduced biodiversity in the urban area to the presence of an excess of some forms of N, NH4+ in particular. However, it does not consider other pollutant that could have influenced microbial communities such as heavy metals or other chemical pollutants, probably present.
Two analyzes are missing which could be very indicative and easily detectable, the BOD and the COD. These analyzes give an interesting indication if the DO is influenced by the biological component or give an indication of a possible pollution present.

From the results of the NGS it appears that the nitrifying bacteria (by the way they are chemoautotrophic bacteria, not heterotrophic, line 329) are more abundant in the suburban areas than the urban. The possibility, as claimed in the discussion (lines 329-332) that they are negatively influenced by the excess of ammonium, does not seem likely to me. I think there may be other pollutants that block their activity, such as heavy metals. Nitrifying bacteria can work with very high NH4+ amount, as they are active in urban wastewater treatment plants. However, they need oxygen to oxidize ammonium – in the water of the sampling site S2, for example, where DO is high, there is almost no NH4+.

English language needs some revision.

Some suggestions and notes in particular:

Line 40: why only N2O and not NOX in general?

Line 73: why “urgently”?

Line 85: “in the river sediments of highly urbanized city, Shanghai”

Line 86: “to examine that the shifts in..” -> to examine whether the shifts…

Line 87: “to reveal that what variables played..” -> to study which variables..

Line 98: “sites were located at central urban area” -> sites were located at the central urban area.

Figure 4: it is useful if you add next to the different point  also which site they refer to

Author Response

Response to Reviewer 1 Comments

Point 1: The study, in fact, ascribe the reduced biodiversity in the urban area to the presence of an excess of some forms of N, NH4+ in particular. However, it does not consider other pollutant that could have influenced microbial communities such as heavy metals or other chemical pollutants, probably present.

Response 1: Yes, heavy metals or other chemical pollutants also contributed to the total variation in bacterial community composition as reported in previous studies. This effect has also been clarified in the discussion section.

Point 2: Two analyzes are missing which could be very indicative and easily detectable, the BOD and the COD. These analyzes give an interesting indication if the DO is influenced by the biological component or give an indication of a possible pollution present.

Response 2: Yes, DO is the result of BOD and COD, and thus affects the biogeochemical processes. Large pollutant inputs, especially the nitrogen in the rivers will cause large oxygen consumption, leading to a lower DO. Therefore, the DO could be an important factors affecting the biological processes.

Point 3: From the results of the NGS it appears that the nitrifying bacteria (by the way they are chemoautotrophic bacteria, not heterotrophic, line 329) are more abundant in the suburban areas than the urban. The possibility, as claimed in the discussion (lines 329-332) that they are negatively influenced by the excess of ammonium, does not seem likely to me. I think there may be other pollutants that block their activity, such as heavy metals. Nitrifying bacteria can work with very high NH4+ amount, as they are active in urban wastewater treatment plants. However, they need oxygen to oxidize ammonium – in the water of the sampling site S2, for example, where DO is high, there is almost no NH4+.

Response 3: Yes, the nitrifying bacteria are generally chemoautotrophic bacteria. Although the NH4+ contents (6.4–15.4 mg N L–1) are extremely high in the urban rivers, nitrifying bacteria cannot be directly inhibited by the toxicity of the NH4+ in this study due to that nitrifying bacteria can work with high NH4+ amount (up to 196 mg N L–1). Thus, the attribute in this result may be that the nitrifiers were inhibited by low DO concentration in the urban rivers, which was induced by NH4+. In addition, this result may also be attributed to other pollutants, such as heavy metals and permanent organic pollutants, as reported in previous studies in urban river ecosystems.

Point 4: Line 40: why only N2O and not NOX- in general?

Response 4: It has been revised.

Point 5: Line 73: why “urgently”?

Response 5: It has been revised.

Point 6: Line 85: “in the river sediments of highly urbanized city, Shanghai”

Response 6: It has been revised.

Point 7: Line 86: “to examine that the shifts in.” -> to examine whether the shifts…

Response 7: It has been revised.

Point 8: Line 87: “to reveal that what variables played.” -> to study which variables..

Response 8: It has been revised.

Point 9: Line 98: “sites were located at central urban area” -> sites were located at the central urban area.

Response 9: It has been revised.

Point 10: Figure 4: it is useful if you add next to the different point also which site they refer to

Response 10: It has been revised in Figure 4.

Reviewer 2 Report

The manuscript entitled “Bacterial community shifts driven by nitrogen pollution in river sediments of a highly urbanized city” by Lin and cols. analyze water quality of Shanghai rivers in terms of their chemical properties and microbiome richness and diversity, and study the relationship of these parameters. They find that urban rivers of Shanghai are really contaminated. High pollution of urban river sediments negatively correlates with the abundance of Firmicutesand Nitrospirae, and the increase phylum Bacteroidetes, Verrucomicrobia and Spirochaetes. These findings confirm  that human activity influences drastically microbial ecology of aquatic environments, and are of special relevance for the region of Shanghai. These findings are of special relevance and deserve be published.

I only have an important concern that should be addressed before acceptance of the manuscript. In the present form, the manuscript does not reach the basic English level that should be presented in a serious scientific journal as ijerph is. There are so may language and grammar mistakes and phrases that are incomprehensible. I started with a list of “minor points”, but after several paragraphs I decided recommend the authors a substantial revision of grammar. I give some examples below.

Line 18: Sediment NO3– and NH4+ were highly accumulated àsediments containing NO3 and NH4+…

Line 21: Bacterial communitiesProteobacteria, Firmicutes, and Bacteroidetes were the dominated bacterial phylum communities

Line 35: has a negative influence

Line 38: environmental problems

Line 41: Therefore,thestudies regarding thenitrogen pollution and the associated ecological effects have been raised for decades.

Line 47: It has been suggested that environmental properties are the important factors defining the microbial richness and diversity. You mean: It has been suggested that environmental factors affect microbial richness and diversity in ecosystems?

Author Response

Response to Reviewer 2 Comments

Point 1: I only have an important concern that should be addressed before acceptance of the manuscript. In the present form, the manuscript does not reach the basic English level that should be presented in a serious scientific journal as ijerph is. There are so may language and grammar mistakes and phrases that are incomprehensible. I started with a list of “minor points”, but after several paragraphs I decided recommend the authors a substantial revision of grammar. I give some examples below.

Response 1: Thank you for this suggestion. We have improved the English level of this manuscript.

Point 2: Line 18: Sediment NO3 and NH4+ were highly accumulated sediments containing NO3 and NH4+

Response 2: It has been revised.

Point 3: Line 21: Bacterial communities Proteobacteria, Firmicutes, and Bacteroidetes were the dominated bacterial phylum communities

Response 3: It has been revised.

Point 4: Line 35: has a negative influence

Response 4: It has been revised.

Point 5: Line 38: environmental problems

Response 5: It has been revised.

Point 6: Line 41: Therefore, the studies regarding the nitrogen pollution and the associated ecological effects have been raised for decades.

Response 6: It has been revised.

Point 7: Line 47: It has been suggested that environmental properties are the important factors defining the microbial richness and diversity. You mean: It has been suggested that environmental factors affect microbial richness and diversity in ecosystems?

Response 7: It has been revised.

Reviewer 3 Report

Dear Authors,

The present paper spans important microbial aspects of nitrogen pollution in the river sediments of a highly urbanized city. I liked the context and organization of the article submitted for the journal. The paper is well constructed and very well written. I can see that the authors gave a significant effort to make the paper well written and compelling.

There is enough background information in the introduction section with clearly stated objectives of the research. The materials and method section addressed all the scientific techniques followed during the experiment. The discussion section has some well-argued points with some good recent article. The conclusion also supports the results and address the objectives of the manuscript.

However, some edits and/or clarifications needed for some of the statistical tests done (please see the comments below).

Regards,  

Comments:

Line 250-252:

Couldn't find any line about ANOSIM in the statistical analysis section.
Please include a sentence about ANOSIM in the statistical analysis section.
Also, please provide ANOSIM results in the supplementary data.

Line 258 and figure 4:

What was the stress value for NMDS analysis?
Please mention the stress value either in the NMDS plot or elsewhere in the text.

Author Response

Response to Reviewer 3 Comments

Point 1: Line 250-252: Couldn't find any line about ANOSIM in the statistical analysis section. Please include a sentence about ANOSIM in the statistical analysis section. Also, please provide ANOSIM results in the supplementary data.

Response 1: It has been revised.

Point 2: Line 258 and figure 4: What was the stress value for NMDS analysis? Please mention the stress value either in the NMDS plot or elsewhere in the text.

Response 2: The stress value represents the difference between the distance between a point in a two-dimensional space and a point in a multidimensional space. The value has been added in the text.

Round 2

Reviewer 1 Report

The manuscript was revised, the authors have sufficiently answered the questions asked and it can be published.